# The prevalence and risk of mortality associated with intradialytic hypertension among patients with end-stage kidney disease on haemodialysis: A systematic review and meta-analysis

Oluseyi Ademola Adejumo[1], Imuetinyan Rahsida Edeki[2], Dapo Sunday Oyedepo[3], Olawale Elijah Yisau[1], Olanrewaju Olumide Ige[1], Inyeneabasi Udeme Ekrikpo[4], Ayman Sabri Moussa[5], Hansel Palencia[6], Jean Jacques Noubiap[7], Udeme Ekpenyong Ekrikpo[5,8]*

1 Department of Internal Medicine, University of Medical Sciences, Ondo State, Nigeria, 2 Department of Internal Medicine, University of Benin Teaching Hospital, Edo State, Nigeria, 3 Department of Internal Medicine, University of Ilorin Teaching Hospital, Kwara State, Nigeria, 4 Department of Biology, Western Kentucky University, Bowling Green, Kentucky, United States of America, 5 Research Team, DaVita HealthCare, Riyadh, Saudi Arabia, 6 International Clinical Team, DaVita International, London, United Kingdom, 7 Division of Cardiology, Department of Medicine, University of California - San Francisco, San Francisco, California, United States of America, 8 Department of Internal Medicine, University of Uyo, Akwa Ibom State, Nigeria

* udemeekrikpo@uniuyo.edu.ng

## Abstract

### Introduction

Intradialytic hypertension (IDHTN) is a common but less frequently recognised complication of haemodialysis. However, it is associated with increased overall mortality in patients on haemodialysis. This systematic review and meta-analysis aimed to determine the prevalence of IDHTN and associated mortality risk in the global haemodialysis population.

### Method

A systematic search of PubMed and EMBASE was undertaken to identify articles with relevant data published between 1990 and 2023. The pooled prevalence of IDHTN in the global haemodialysis population was determined using the DerSimonian-Laird random-effects meta-analysis. The pooled hazards ratio for mortality in patients with IDHTN was also computed from the studies that reported mortality among haemodialysis patients with IDHTN. The study protocol was registered with PROSPERO (CRD42023388278).

### Results

Thirty-two articles from 17 countries were included, with a pooled population of 127,080 hemodialysis patients (median age 55.1 years, 38.2% females). Most studies had medium methodological quality (53.1%, n = 17). The overall pooled prevalence of IDHTN was 26.6%

**Data Availability Statement:** All relevant data are within the manuscript and its Supporting information files.

**Funding:** The author(s) received no specific funding for this work.

**Competing interests:** The authors have declared that no competing interests exist.

[(95% CI 20.2–33.4%), n = 27 studies, $I^2$ = 99.3%, p<0.001 for heterogeneity], with significant differences depending on the definition used. The pooled proportion of haemodialysis sessions with IDHTN was 19.9% [(95% 12.5–28.6%, n = 8 studies, $I^2$ = 99.3%, p<0.001 for heterogeneity)] with significant differences across the different definition criteria. The p-value for the Begg test was 0.85. The median pre-dialysis blood pressure was not significantly associated with IDHTN. The pooled hazard ratio for mortality was 1.37 (95% CI 1.09–1.65), n = 5 studies, $I^2$ = 13.7%, and p-value for heterogeneity = 0.33.

## Conclusion

The prevalence of IDHTN is high and varies widely according to the definition used. A consensus definition of IDHTN is needed to promote uniformity in research and management. The increased mortality risk forecasted by IDHTN highlights the need for optimal blood pressure control in patients on hemodialysis.

## Introduction

The number of patients with CKD has steadily increased in recent years. In 2017, 697.5 million people were reported to be affected by CKD, representing about 10% of the global population [1]. As the disease progresses to end stage, patients require kidney replacement therapy (KRT) to stay alive. Although kidney transplantation is the most preferred form of KRT, the majority of patients have to be on dialysis before eventually undergoing a kidney transplant [2] if found eligible. Haemodialysis is thus the most common form of KRT globally and is associated with complications that may be life-threatening [2].

Blood pressure fluctuation is frequently experienced by patients undergoing haemodialysis. This makes them highly susceptible to intradialytic hypotension and hypertension during haemodialysis. Intradialytic hypertension (IDHTN) is one of the most commonly encountered complications of haemodialysis [3–7]. IDHTN tends to be more common in older individuals and those with lower dry weight, lower interdialytic weight gain, lower serum creatinine and albumin, and those on multiple antihypertensive medications [3, 4]. Several factors have been implicated in the pathophysiology of IDHTN. These include volume overload, increased sympathetic and renin activation, endothelin release, intradialytic sodium gain, electrolyte imbalance, use of high calcium dialysate, intravenous use of erythropoiesis-stimulating agents and removal of antihypertensive medication during dialysis [8, 9].

Although there are no unified criteria for diagnosis of IDHTN, most studies defined it as an increase in systolic blood pressure of >10 mmHg or a 15 mmHg rise in mean arterial pressure during dialysis or immediately after dialysis [10–12]. It is often neglected, and its frequency and intervention are not discussed as often as intradialytic hypotension. Some studies have suggested that IDHTN might be associated with an increased risk of mortality [12, 13]. However, the data on the magnitude and adverse effects of IDHTN have not been summarised. Therefore, this study aimed to determine the prevalence and risk of mortality associated with IDHTN among patients with end-stage kidney disease (ESRD) on haemodialysis [8, 9].

## Methods

This study was registered with PROSPERO (registration number CRD42023388278) and reported according to the Preferred Reporting Items for Systematic Review and Meta-Analysis (PRISMA) guideline [14].

MEDLINE/PubMed and EMBASE (S2 Table) were searched to identify relevant studies. Two researchers independently selected the studies using predetermined inclusion and exclusion criteria. There were no language, geographical locations, study design or sample size restrictions. The search of the articles was done between March and June 2023. We included studies published between 1st January 1990 and 30th June 2023 that used various definitions for IDHTN, including an increase in mean arterial pressure >15mmHg within or immediately post dialysis [5]; greater than or equal to 10mmHg increase in systolic blood pressure during or at the end of hemodialysis [4]; rise in systolic blood pressure greater than 10mmHg from pre to post-dialysis in an average of 3 consecutive dialysis sessions [15]; blood pressure rise of any degree during the second or third intradialytic hour [3]; hypertension that appears resistant to ultrafiltration and which occurs during or immediately after hemodialysis [16]; systolic blood pressure rise greater than 10mmHg from pre- to post-dialysis in the hypertensive range in at least 4 out of 6 consecutive dialysis sessions [17]. For ease of analysis, the definitions of IDHTN were categorized into five as follows: Category A (Increase of SBP by 10mmHg or more during or at the end of the hemodialysis session); Category B (Increased SBP by 10mmHg or more in 3 consecutive hemodialysis sessions); Category C (Increased SBP by 10mmHg or more in at least 4 out of 6 hemodialysis sessions); Category D (Increased SBP by 10mmHg or increased mean arterial blood pressure (MABP) by 15mmHg during or at the end of the dialysis session); and E (Increased MABP by 15mmHg or more during or at the end of hemodialysis). Studies where the prevalence of IDHTN or criteria for IDHTN definition was not stated were excluded.

Two reviewers (OAA and IRE) independently screened the titles and abstracts of identified citations and then performed a detailed review of all selected full texts to ascertain eligibility. Disagreements were resolved through discussion and consensus. The following variables were extracted from selected studies: the last name of the first author, year of publication, country and continent where the study was done, the sample size of the study, duration of the study, study design, mean age of the study participants, percentage of the study population that was female, percentage of dialysis patients with diabetes mellitus as the primary cause of kidney failure, the total number of haemodialysis sessions, number of patients or haemodialysis sessions with IDHTN, mean pre-dialysis and post-dialysis systolic and diastolic blood pressure, intradialytic weight gain, ultrafiltration goal, definition criteria for IDHTN, serum sodium and bicarbonate, dialysate electrolytes, use of IV erythropoietin and type of antihypertensive medications used by the patients and hazard ratio for mortality of patients with IDHTN.

The Joanna-Briggs Institute Critical Appraisal Checklist for Studies Reporting Prevalence Data was used to assess the methodological quality of the constituent studies [18]. Studies scored 1 for each of the nine questions with a "yes" response. Studies with a cumulative score between 0 and 3 were regarded as poor quality, 4 to 6 as intermediate or medium quality, and 7 to 9 as high quality.

Stata 18.0 (Stata Corp., 2023. Stata Statistical Software: Release 18, College Station, TX) was used for statistical analysis. The pooled prevalence of IDHTN in the global haemodialysis population was determined using random effects meta-analytic techniques. The study-specific estimates derived from the DerSimonian-Laird random effects model [19] were pooled to estimate the prevalence of IDHTN in this population. To minimise the effect of extreme values, the Freeman-Tukey double arcsine transformation [20] was used to stabilise the individual study variances before using the random effects model to obtain the pooled estimates. Publication bias was assessed using the Begg test [21]. A sub-group analysis was undertaken to compare summary estimates based on IDHTN definition, presence of diabetes mellitus, and age. The median values of age, pre-dialysis blood pressure, and proportion of patients with diabetes mellitus were used as the discriminatory value for grouping studies for subgroup analysis. The

$I^2$ statistic was used to determine the between-study heterogeneity, with 25%, 50%, and 75% representing the upper thresholds for low, medium and high heterogeneity [22]. A leave-one-out meta-analysis was used as sensitivity analysis.

## Results

### Study selection and characteristics

The systematic literature search identified 22,720 articles from PubMed, EMBASE, and hand searching (S2 Table). After duplicate removal, title and abstract screening, and full article assessment for eligibility, 32 articles [3, 6, 12, 13, 23–50] were included (Fig 1). The publication years of the included articles ranged from 2007 to 2023, with a pooled population of 127,080 ESKD patients on hemodialysis from 17 countries. There were 847 participants (6 studies) from Africa, 5105 (4 studies) from Europe, 119,050 (6 studies) from North America, and 2078 (16 studies) from Asia. Table 1 is a summary of data extracted from the included articles. Most (71.9%) of the studies were cross-sectional in design. One study [3] was a secondary analysis of a clinical trial, while eight (25.0%) of the studies were cohort studies.

The eight definitions of IDHTN used in the constituent studies were categorised into five groups based on the magnitude of blood pressure increase, frequency of occurrence, and if systolic blood pressure (SBP) or mean arterial blood pressure (MABP) was used in the definition (S3 Table).

The sample size of the component studies ranged from 28 [47] to 113,255 patients [42]. Twenty-eight (87.5%) of the studies reported IDHTN frequency as a proportion of the total number of patients in the study, while eight studies (25.0%) [6, 27, 30, 37, 38, 41, 44, 47] reported it as a proportion of the total number of sessions. Three studies [37, 44, 47] reported both the total number of patients and the total number of sessions. The mean age of the participants ranged from 43.4 years [38] to 70.0 years [32], with a median female proportion of 38.2% (32.7–42.8%). The median proportion of patients with diabetes mellitus was 39.5% (interquartile range 18.3–50.0%), with a range of 7.6% to 73.0%. Five studies [3, 12, 29, 34, 39] reported adjusted hazard ratios for mortality for patients with intradialytic hypertension.

Most of the studies had medium methodological quality (53.1%, n = 17 studies) (Table 1), 6 studies (18.7%) were of high quality, while nine studies (28.1%) had poor methodological quality. Table 1 provides a summary of data extracted from the constituent articles.

### Prevalence of IDHTN

The overall pooled prevalence of IDHTN was 26.6% [(95% CI 20.2–33.4%), n = 27 studies, $I^2$ = 99.3%, p<0.001 for heterogeneity] irrespective of the definition used. Studies using definitions A, B, C, and E had pooled prevalence of 22.9% (15.3–31.4%), 27.0% (13.5–43.0%), 37.0% (24.8–50.1%), and 18.5% (14.1–23.3%) respectively (Fig 2). Only one study [25] used definition D and had data to analyse the proportion of patients with IDHTN.

The pooled proportion of sessions with IDHTN was 19.9% [(95% 12.5–28.6%, n = 8 studies, $I^2$ = 99.3%, p<0.001 for heterogeneity)] with significant differences across the different definition criteria (S1 Fig). The p-value for the Begg test was 0.85, and the corresponding funnel plot corroborates the suggestion of no publication bias (S2 Fig).

There was no significant difference in the pooled prevalence of IDHTN for studies with median pre-dialysis SBP greater than 147.2mmHg [29.4% (95% CI 19.5–40.4%)] compared to the pooled prevalence of studies with pre-dialysis SBP less than or equal to 147.2mmHg [21.8% (17.4–26.5%), p = 0.18], S3 Fig. There was no difference in pooled prevalence between studies with a high proportion of diabetes mellitus patients compared to those with a lower proportion of diabetes mellitus patients [27.6% (95% CI 19.4–36.6) versus 25.8% (95%CI

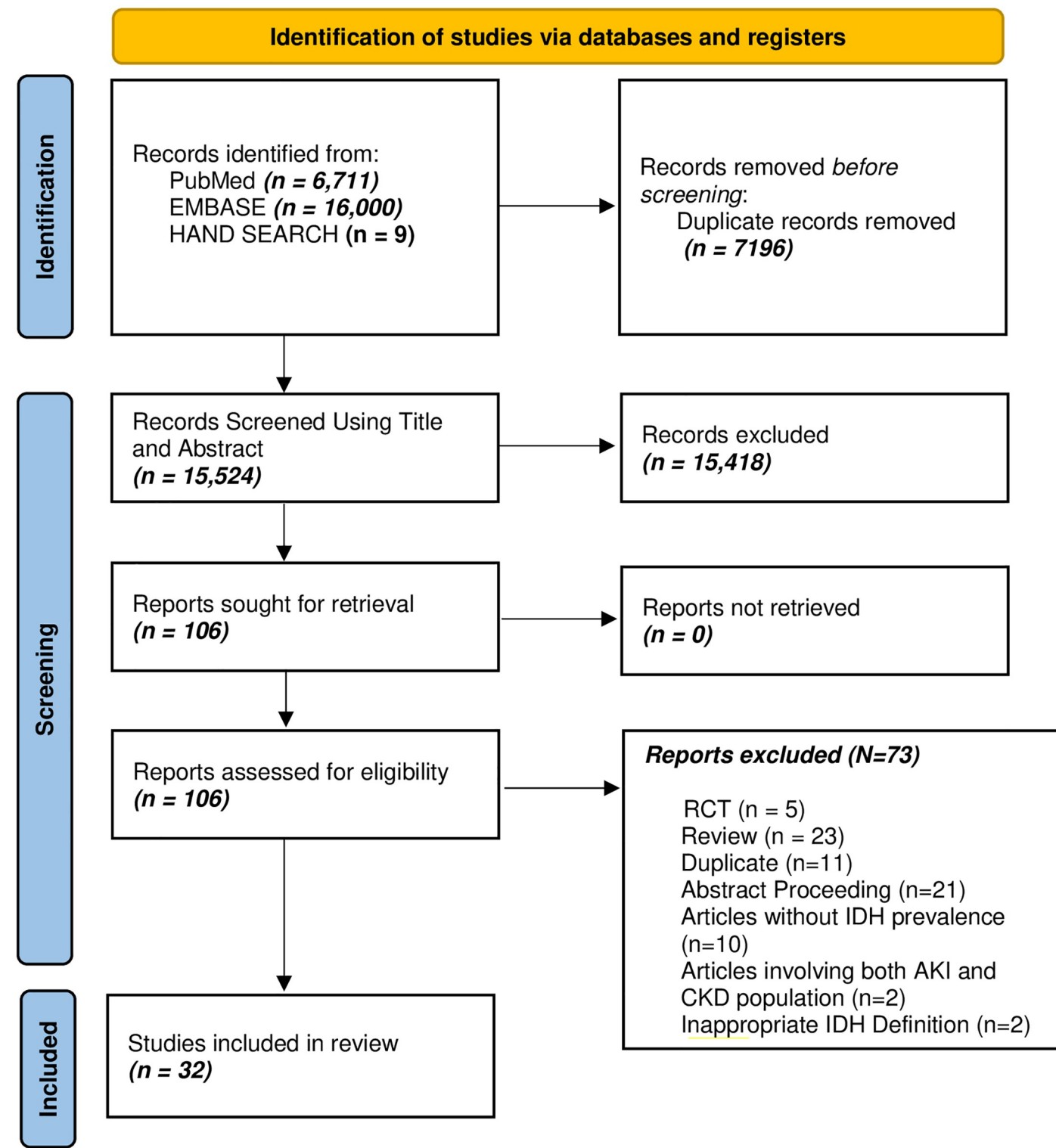

**Fig 1. PRISMA flow chart—Identification and screening of articles.**

**Table 1. Summary of extracted data.**

| Author Name | Publication Year | Country | Continent | Study Type | Study Duration (months) | Mean age | Female (%) | Sample size | Total No. of sessions | Number of IDHTN patients | No. of IDHTN sessions | Diagnostic Criteria | Mean Pre-HD BP | Mean Post-HD BP | Dialysis vintage (months) | JBI Score |
|---|---|---|---|---|---|---|---|---|---|---|---|---|---|---|---|---|
| Inrig [34] | 2007 | United States | North America | CT | 6 | 58.5 | 48.9 | 438 | - | 58 | - | Increased SBP by 10mmHg or more during or end of HD | 150/80 | 138/74 | - | 3 |
| Inrig [3] | 2009 | United States | North America | C | 24 | 61.6 | 48.5 | 1748 | - | 213 | - | Increased SBP by 10mmHg or more during or end of HD | 151/78 | 142/75 | - | 5 |
| Rubinger [37] | 2012 | Israel | Asia | CS | - | 58 | 36.1 | 108 | 113 | 57 | 62 | Increased SBP by 10mmHg or more during or end of HD | 135/67 | 134/73 | 22.8 | 4 |
| Amira [6] | 2012 | Nigeria | Africa | CS | 12 | 47.5 | 43.8 | 201 | 1010 | - | 154 | Any Increase in BP during 2nd or 3rd hour of HD; Any increase in BP that is resistant to UF during HD | 151/91 | 157/90 | 2.4 | 5 |
| Van Buren [44] | 2012 | United States | North America | C | 7 | 54.3 | 41.0 | 362 | 22955 | 29 | 4889 | Increased SBP by 10mmHg or more during or end of HD | 151/82 | 139/77 | - | 5 |
| Agrawal [47] | 2012 | Nepal | Asia | C | 6 | 48.8 | 32.0 | 28 | 1455 | 14 | 58 | Increased SBP by 10mmHg or more in at least 4 out of 6 HD sessions | - | - | - | 2 |
| Oosugi [32] | 2013 | Japan | Asia | C | 41 | 70.0 | 56.0 | 84 | 672 | 21 | - | Increased SBP by 10mmHg or more during or end of HD | 138/75 | 168/82 | 54.0 | 2 |
| Park [42] | 2013 | United States | North America | C | 60 | 61.0 | 44.9 | 113255 | - | 11994 | - | Increased SBP by 10mmHg or more during or end of HD | 149/77 | 140/72 | 3.0 | 6 |
| Movilli [43] | 2013 | Italy | Europe | C | 8 | 68.0 | 37.4 | 206 | 4944 | 35 | - | Any Increase in BP during 2nd or 3rd hour of HD | 139/72 | 134/71 | 37.0 | 6 |
| Nongnuch [13] | 2015 | United Kingdom | Europe | C | | 60.3 | 38.0 | 531 | - | 96 | - | Increased SBP by 10mmHg or more during or end of HD | 141/72 | 130/72 | 42.0 | 4 |

(*Continued*)

Table 1. (Continued)

| Author Name | Publication Year | Country | Continent | Study Type | Study Duration (months) | Mean age | Female (%) | Sample size | Total No. of sessions | Number of IDHTN patients | No. of IDHTN sessions | Diagnostic Criteria | Mean Pre-HD BP | Mean Post-HD BP | Dialysis vintage (months) | JBI Score |
|---|---|---|---|---|---|---|---|---|---|---|---|---|---|---|---|---|
| Sebastian [28] | 2016 | South Africa | Africa | C | 18 | 55.7 | 42.0 | 190 | - | 54 | - | Increased SBP by 10mmHg or more in at least 4 out of 6 HD sessions | 153.1/- | - | - | 7 |
| Losito [39] | 2015 | Italy | Europe | C | 1 | 65.1 | 38.0 | 4292 | 51504 | 994 | - | Increased SBP by 10mmHg or more during or end of HD | 136.3/- | - | 67.5 | 5 |
| Diaz [46] | 2016 | Puerto Rico | North America | C | 6 | - | 32.7 | 49 | - | 8 | - | Increased SBP by 10mmHg or more during or end of HD | - | - | - | 6 |
| Ren [31] | 2017 | China | Asia | C | 3 | 49.4 | 38.9 | 131 | - | 14 | - | Increased SBP by 10mmHg or more during or end of HD | 147/88 | - | 68.0 | 5 |
| Nilrohit [35] | 2017 | India | Asia | C | 30 | 61.0 | 27.5 | 142 | - | 49 | - | Increased SBP by 10mmHg or more from 3 consecutive HD sessions | - | - | - | 5 |
| Islam [38] | 2017 | Pakistan | Asia | C | 3 | 43.4 | 16.7 | 150 | 2520 | - | 208 | Increased MABP by 15mmHg or more during or end of HD; Increased SBP by 10mmHg or more during or end of HD | - | - | - | 3 |
| Choi [29] | 2018 | Korea, South | Asia | C | 36 | 54.0 | 42.5 | 73 | - | 14 | - | Increased SBP by 10mmHg or more from 3 consecutive HD sessions | 140/80 | 134/79 | - | 6 |
| Raikou [36] | 2018 | Greece | Europe | C | 60 | 62.2 | 38.2 | 76 | - | 15 | - | Increased SBP by 10mmHg or more during or end of HD | - | - | 45.2 | 3 |
| Mahmood [40] | 2018 | Pakistan | Asia | C | 7 | 56.2 | 33.0 | 100 | - | 10 | - | Increased SBP by 10mmHg or more during or end of HD | - | - | - | 3 |

(Continued)

**Table 1.** (Continued)

| Author Name | Publication Year | Country | Continent | Study Type | Study Duration (months) | Mean age | Female (%) | Sample size | Total No. of sessions | Number of IDHTN patients | No. of IDHTN sessions | Diagnostic Criteria | Mean Pre-HD BP | Mean Post-HD BP | Dialysis vintage (months) | JBI Score |
|---|---|---|---|---|---|---|---|---|---|---|---|---|---|---|---|---|
| Labarcon [50] | 2018 | Philippines | Asia | C | 3 | 45.0 | 41.8 | 309 | - | 116 | - | Increased SBP by 10mmHg or more in at least 4 out of 6 HD sessions | - | - | 40.8 | 4 |
| Okpa [26] | 2019 | Nigeria | Africa | C | 24 | 51.7 | 40.6 | 64 | - | 29 | - | Increased MABP by 15mmHg or more during or end of HD; Increased SBP by 10mmHg or more during or end of HD | 160/91 | 160/90 | - | 6 |
| Veerappan [27] | 2019 | India | Asia | C | 12 | 50.6 | 27.0 | 60 | 240 | - | 123 | Increased MABP by 15mmHg or more during or end of HD; Increased SBP by 10mmHg or more during or end of HD | - | - | 8.0 | 7 |
| Liu [24] | 2020 | China | Asia | C | 3 | 64.3 | 45.8 | 144 | 5616 | 34 | - | Increased SBP by 10mmHg or more during or end of HD | 143/72 | 145/71 | 33.5 | 6 |
| Raja [30] | 2020 | Eritrea | Africa | C | 5 | 53.0 | 34.5 | 29 | 573 | - | 29 | Increased MABP by 15mmHg or more during or end of HD | - | - | - | 5 |
| Diakite [33] | 2020 | Guinea | Africa | C | 3 | 45.5 | - | 131 | - | 53 | - | Increased SBP by 10mmHg or more in at least 4 out of 6 HD sessions | 148/88 | - | 23.51 | 4 |
| Kale [45] | 2020 | India | Asia | C | 23 | 55.1 | 30.8 | 91 | - | 20 | - | Any Increase in BP during 2nd or 3rd hour of HD; Increased SBP by 10mmHg or more from 3 consecutive HD sessions | 147/75 | 151/78 | 47.0 | 4 |

*(Continued)*

Table 1. (Continued)

| Author Name | Publication Year | Country | Continent | Study Type | Study Duration (months) | Mean age | Female (%) | Sample size | Total No. of sessions | Number of IDHTN patients | No. of IDHTN sessions | Diagnostic Criteria | Mean Pre-HD BP | Mean Post-HD BP | Dialysis vintage (months) | JBI Score |
|---|---|---|---|---|---|---|---|---|---|---|---|---|---|---|---|---|
| Nayak [48] | 2020 | India | Asia | C | 3 | - | 24.8 | 165 | - | 135 | - | Increased SBP by 10mmHg or more during or end of HD | - | - | - | 3 |
| Ali [25] | 2021 | Pakistan | Asia | C | 6 | 45.5 | 34.0 | 94 | 5544 | 16 | - | Increased SBP by 10mmHg or more during or end of HD | - | - | 60.0 | 6 |
| Mujtaba [23] | 2022 | Pakistan | Asia | C | 3 | 51.0 | 43.7 | 263 | - | 42 | - | Increased SBP by 10mmHg or more in at least 4 out of 6 HD sessions | 141/76 | 141/76 | - | 4 |
| Singh [12] | 2022 | United States | North America | C | 3 | 62.0 | 42.8 | 3198 | - | 1502 | - | Increased SBP by 10mmHg or more during or end of HD | 149/- | 136/- | 31.0 | 4 |
| Prabhu [49] | 2022 | India | Asia | C | 3 | - | 29.0 | 136 | - | 78 | - | Increased SBP by 10mmHg or more in at least 4 out of 6 HD sessions | - | - | - | 3 |
| Uduagbanen [41] | 2023 | Nigeria | Africa | C | 41 | 49.9 | 38.5 | 232 | 1248 | - | 305 | Increased SBP by 10mmHg or more during or end of HD | - | - | 36.0 | 6 |

JBI = Joanna Briggs Institute; HD = Haemodialysis; BP = Blood pressure; Cross-sectional = CS; Cohort = C, Clinical Trial = CT.

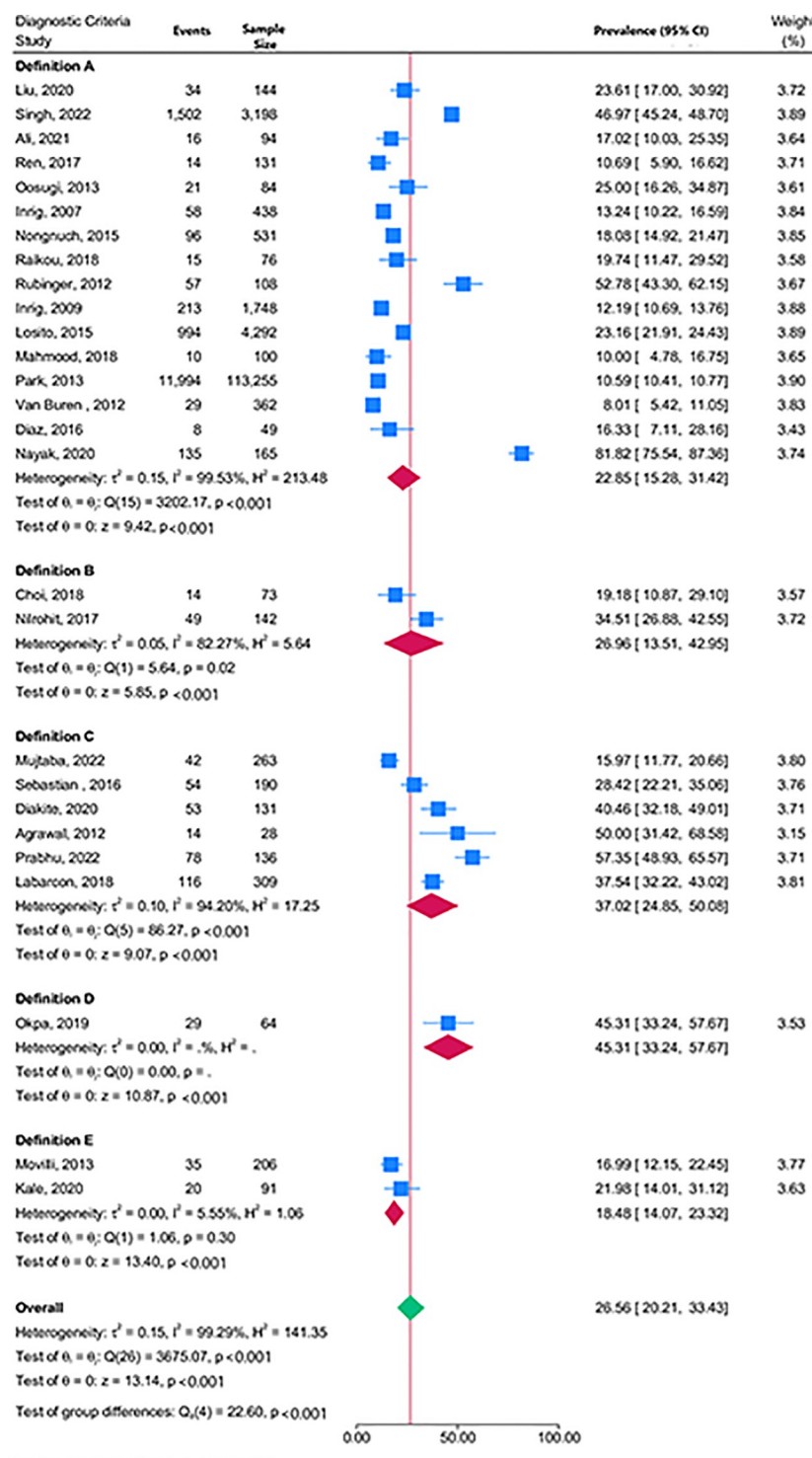

**Fig 2. Forest plot showing the pooled prevalence of IDHTN by definition criteria.**

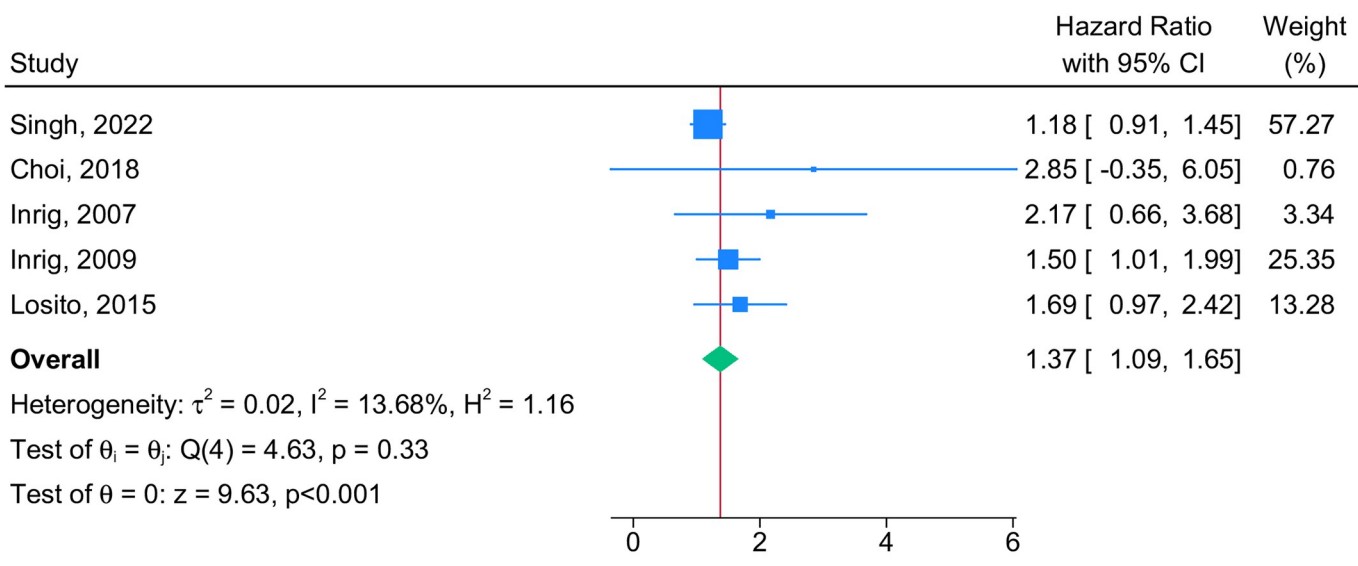

**Fig 3. Forest plot showing the pooled hazard ratio for mortality among IDHTN patients.**

20.5–31.5%), p = 0.82], S4 Fig. There was also no difference in the pooled prevalence of studies with median age less than 65 years compared to those at least 65 years [24.0% (95%CI 16.7% – 32.2%) versus 21.6% (17.4–26.0%), p = 0.58], S5 Fig. The pooled prevalence for IDHTN was highest in Africa [37.2% (27.1–47.8)], followed by Asia [31.4% (20.4–43.6)], Europe [19.8% (16.3–23.7)], and North America [16.6% (5.5–32.1%)], p<0.001 for difference across continents (S6 Fig).

Leave-one-out meta-analysis showed a maximum IDHTN pooled prevalence deviation of 2.0%, seen when Nayak et al. [48] were omitted from the list of pooled studies (S7 Fig).

**Mortality risk.** IDHTN was associated with a 37% increased risk of mortality in patients on hemodialysis [pooled hazard ratio 1.37 (95% CI 1.09–1.65), n = 5 studies, $I^2$ = 13.7%, p = 0.33, Fig 3].

## Discussion

This systematic review and meta-analysis of 32 studies determined the pooled prevalence of IDHTN among 127,080 ESRD patients on maintenance haemodialysis from 17 countries spread across the globe. The pooled prevalence rates of IDHTN among chronic haemodialysis patients and during haemodialysis sessions were 26.6% and 19.9%, respectively, with significant variability depending on the definition of IDHTN. There was a 37% increased mortality risk among haemodialysis patients with IDHTN.

The pooled prevalence of IDHTN is higher than 5–15%, as reported by Dorhout et al. [51] in a report published almost three decades ago. The higher pooled prevalence in this present systematic review and meta-analysis may be due to the relatively better awareness of IDHTN and its consequences now compared to previous decades. In addition, the higher prevalence of IDHTN seen in this present study may be partly because this review considered studies with diverse criteria and studies that combined different criteria. The reason for the higher prevalence of IDHTN among Africans is uncertain but may be related to the established finding of a higher prevalence of hypertension among Blacks compared to other racial groups [52].

A major challenge encountered in the diagnosis of IDHTN is that there is no consensus definition. The Kidney Disease: Improving Global Outcomes (KDIGO) proposed the definition of IDHTN as increased systolic blood pressure by 10 mmHg or more in at least 4 out of 6 hemodialysis sessions [53]. However, only 18.8% of the total articles in this systematic review used the criterion, while more than half used a criterion defined as an increase in systolic blood pressure by 10mmHg or more during or at the end of the hemodialysis session. This observation may be because the latter criterion appears easier to implement than the criterion proposed by KDIGO.

Although the exact pathophysiology of IDHTN is not well understood, different mechanisms have been postulated, such as an increase in sympathetic nervous stimulation, removal of antihypertensive medication during haemodialysis, volume overload, intradialytic sodium gain, intradialytic electrolyte imbalance, activation of the renin-angiotensin-aldosterone system (RAAS), intravenous administration of erythropoietin stimulating agent and endothelial cell dysfunction [8].

IDHTN has not received the desired attention despite the fact that it is common and has various clinical consequences that may impact the overall outcomes of ESRD on haemodialysis. This review showed that patients with IDHTN have a 37% increased mortality rate compared to those without IDHTN. The ambulatory blood pressure pattern in haemodialysis patients showed that the elevated post-dialysis blood pressure in patients with IDHTN persisted for several hours, contributing to an increase in blood pressure burden in this group of patients [54, 55]. IDHTN has been found to be an independent risk factor for left ventricular hypertrophy. Shamir et al. [56] reported that there is a 0.2 g/m$^2$ increase in the left ventricular mass index for every 1 mmHg rise in the systolic blood pressure during haemodialysis. Left ventricular hypertrophy is associated with increased mortality from cardiovascular events in ESRD patients [57]. IDHTN is associated with higher mortality and higher incidence of future cardiovascular events such as stroke and myocardial infarction [11]. It is also associated with increased cardiovascular mortality and all-cause mortality in kidney failure [3, 12, 29, 39, 42]. Reports from the CLIMB study showed that those with IDHTN have a 2.2-fold increased risk of all-cause mortality and hospitalisation. The United States Renal Data System Dialysis Morbidity and Mortality Wave II report showed that an increase of 10mmHg in the systolic blood pressure following haemodialysis is associated with a 12% increased risk of all-cause mortality [3].

A pragmatic approach to managing IDHTN is to optimise blood pressure among chronic haemodialysis patients because of the relationship between IDHTN and ambulatory blood pressure. This is also buttressed by the finding of this study, which showed that those who have elevated pre-dialysis hypertension were more likely to have IDHTN, though the difference did not achieve statistical significance. Interventions that are patient-specific should also be instituted. Based on the possible mechanisms behind IDHTN, the following interventions may be likely beneficial: optimising volume and sodium control, individualised sodium profiling, facilitation of dry weight achievement, optimal dialysis with increased frequency and extended time haemodialysis; inhibition of sympathetic nervous system overactivity by the use of medications that could block both alpha and beta-adrenergic receptors, use of antihypertensive medications including those that block the RAAS that are not easily dialysable, and changing the route of administration of erythropoietin stimulating agents from intravenous to subcutaneous route of administration.

The absence of a standard definition for IDHTN produced significant heterogeneity in the estimates. This was further investigated by performing a subgroup analysis. There was inadequate intradialytic data like dialysate calcium, sodium and potassium concentration and other factors to further investigate their impact on IDHTN prevalence.

## Conclusion

There is a need to have a consensus definition of IDHTH to ensure that the actual magnitude of this condition can be better ascertained. IDHTN is associated with an increased risk of mortality. Therefore, physicians should make more effort toward achieving optimal ambulatory blood pressure control in haemodialysis patients. In addition, patient-specific interventions before and during dialysis should be instituted for those with recurrent IDHTN.

## Supporting information

**S1 Table. PRISMA checklist.**
(DOCX)

**S2 Table. Search terms in PubMed and EMBASE.**
(DOCX)

**S3 Table. IDHTN definition criteria.**
(DOCX)

**S1 Fig. Forest plot showing the pooled prevalence of IDHTN by dialysis session.**
(TIFF)

**S2 Fig. Funnel plot investigating publication bias.**
(TIF)

**S3 Fig. Forest plot comparing the pooled prevalence by pre-dialysis blood pressure.**
(TIFF)

**S4 Fig. Forest plot comparing the pooled prevalence by diabetic status.**
(TIFF)

**S5 Fig. Forest plot comparing the pooled prevalence by age group.**
(TIFF)

**S6 Fig. Forest plot comparing the pooled prevalence by continent.**
(TIFF)

**S7 Fig. Sensitivity analysis using the leave-one-out meta-analysis.**
(TIFF)

## Author Contributions

**Conceptualization:** Oluseyi Ademola Adejumo, Imuetinyan Rahsida Edeki, Dapo Sunday Oyedepo, Olawale Elijah Yisau, Olanrewaju Olumide Ige, Udeme Ekpenyong Ekrikpo.

**Data curation:** Oluseyi Ademola Adejumo, Imuetinyan Rahsida Edeki, Dapo Sunday Oyedepo, Olawale Elijah Yisau, Olanrewaju Olumide Ige, Udeme Ekpenyong Ekrikpo.

**Formal analysis:** Udeme Ekpenyong Ekrikpo.

**Investigation:** Oluseyi Ademola Adejumo, Udeme Ekpenyong Ekrikpo.

**Methodology:** Oluseyi Ademola Adejumo, Imuetinyan Rahsida Edeki, Dapo Sunday Oyedepo, Olawale Elijah Yisau, Olanrewaju Olumide Ige, Inyeneabasi Udeme Ekrikpo, Ayman Sabri Moussa, Hansel Palencia, Jean Jacques Noubiap, Udeme Ekpenyong Ekrikpo.

**Project administration:** Oluseyi Ademola Adejumo, Udeme Ekpenyong Ekrikpo.

**Supervision:** Oluseyi Ademola Adejumo, Udeme Ekpenyong Ekrikpo.

**Validation:** Oluseyi Ademola Adejumo, Ayman Sabri Moussa, Hansel Palencia, Jean Jacques Noubiap, Udeme Ekpenyong Ekrikpo.

**Visualization:** Inyeneabasi Udeme Ekrikpo.

**Writing – original draft:** Oluseyi Ademola Adejumo, Udeme Ekpenyong Ekrikpo.

**Writing – review & editing:** Oluseyi Ademola Adejumo, Imuetinyan Rahsida Edeki, Dapo Sunday Oyedepo, Olawale Elijah Yisau, Olanrewaju Olumide Ige, Inyeneabasi Udeme Ekrikpo, Ayman Sabri Moussa, Hansel Palencia, Jean Jacques Noubiap, Udeme Ekpenyong Ekrikpo.

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
