## [Decision Letter · Decision Letter 0]

24 Mar 2024

PONE-D-23-38254THE PREVALENCE AND RISK OF MORTALITY ASSOCIATED WITH INTRADIALYTIC HYPERTENSION AMONG PATIENTS WITH END-STAGE KIDNEY DISEASE ON HAEMODIALYSIS: A SYSTEMATIC REVIEW AND META-ANALYSISPLOS ONE

Dear Dr. Ekrikpo,

Thank you for submitting your manuscript to PLOS ONE. After careful consideration, we feel that it has merit but does not fully meet PLOS ONE’s publication criteria as it currently stands. Therefore, we invite you to submit a revised version of the manuscript that addresses the points raised during the review process.

We look forward to receiving your revised manuscript.

Kind regards,

Ahmet Murt

Academic Editor

PLOS ONE

Journal Requirements:

Additional Editor Comments:

Our impartial reviewers recommended some major revisions for your manuscript. You can find them below.

Reviewers' comments:

Reviewer's Responses to Questions

**Comments to the Author**

1. Is the manuscript technically sound, and do the data support the conclusions?

Reviewer #1: Yes

Reviewer #2: Yes

2. Has the statistical analysis been performed appropriately and rigorously? 

Reviewer #1: I Don't Know

Reviewer #2: Yes

3. Have the authors made all data underlying the findings in their manuscript fully available?

Reviewer #1: Yes

Reviewer #2: Yes

4. Is the manuscript presented in an intelligible fashion and written in standard English?

Reviewer #1: Yes

Reviewer #2: Yes

5. Review Comments to the Author

Reviewer #1: I enjoyed reading this systematic review and meta-analysis about intradialytic hypertension in hemodialysis patients by Udeme et al. It is a well-written paper in fluent English about an underestimated clinical entity.

Strong points of the analysis are a) the inclusion of studies without geographical or language limitations b) the long study period. On the other hand the absence of a single definition of intradialytic hypertension causes confusion, when comparing prevalence rates in trials.

Minor comments

1. The different definitions of intradialytic hypertension are described in page 4, but the authors should add the A, B ,C…matches (it is stated in the supplementary file, but it is important to be clear in the main text too)

2. The fifth definition is suggested by KDIGO. This could be underlined and discussed a bit more.(less prevalence in this definition)

3. The authors should explain why they use the median values of SBP /DBP and age as cut-off values (supplementary files). I believe that it is more clinically relevant to use eg in age >65 years old vs < 65years old

4. Page 7. Ref 51 the correct form is Dorhout et al instead of Mee et al.

Reviewer #2: In this systematic review authors analyzed the incidence of intradialytic hypertension and mortality rates that is attributed to it.

As very well stated by the authors, there's no exact definition for inradialytic hypertension and this makes inclusion of different studies difficulty.

My recommendations:

1- Please give the exact dates for your included studies.

2- please clarify what are the definitions A,B,C,D and E in the text.

3- Refrain from comparing intradialytic hypertension and hypotension. This can't be the objective of this review.

4- More detailed data is necessary. Please dissect more information from the studies for risk factors that result in intradalytic hypertension and discuss widely. The discussion of this review is still poor. If there's no data in the studies for risk factors please add them as limitations.

5- Please revise the English language of your paper.

6. PLOS authors have the option to publish the peer review history of their article (what does this mean?). If published, this will include your full peer review and any attached files.

Reviewer #1: No

Reviewer #2: No

---

## [Author Response · Author response to Decision Letter 0]

5 Apr 2024

Reviewer 1 

1 The different definitions of intradialytic hypertension are described in page 4, but the authors should add the A, B ,C…matches (it is stated in the supplementary file, but it is important to be clear in the main text too). This has been included in the methodology. Page 4, lines 94-100

2 The fifth definition is suggested by KDIGO. This could be underlined and discussed a bit more.(less prevalence in this definition) This observation is noted and has been included in the discussion. Page 7, line 192-197

3 The authors should explain why they use the median values of SBP /DBP and age as cut-off values (supplementary files). I believe that it is more clinically relevant to use eg in age >65 years old vs < 65years old Thank you, again. We have re-categorized the age groups as suggested. Rerun the analysis and made the necessary changes in the result section. Page 6, Lines 169-171.

4 Page 7. Ref 51 the correct form is Dorhout et al instead of Mee et al. Thank you. This has been corrected. Page 7, Line 186.

 Reviewer 2 

1 Please give the exact dates for your included studies.

 The search of the articles were done between March and June, 2023. We included studies published between 1st January 1990 and 30th June 2023. Page 4. Line 89-90

2 please clarify what are the definitions A,B,C,D and E in the text This has been included in the methodology. P4 lines, line 98-103

3 Refrain from comparing intradialytic hypertension and hypotension. This can't be the objective of this review The discussion on Intradialytic hypotension has been deleted. 

4 More detailed data is necessary. Please dissect more information from the studies for risk factors that result in intradalytic hypertension and discuss widely. The discussion of this review is still poor. If there's no data in the studies for risk factors, please add them as limitations. We do not have data for the risk factors for IDHTN, hence we have included this as part of limitation. 

5 Please revise the English language of your paper The English language has been revised to improve its readability.

---

## [Decision Letter · Decision Letter 1]

16 May 2024

THE PREVALENCE AND RISK OF MORTALITY ASSOCIATED WITH INTRADIALYTIC HYPERTENSION AMONG PATIENTS WITH END-STAGE KIDNEY DISEASE ON HAEMODIALYSIS: A SYSTEMATIC REVIEW AND META-ANALYSIS

PONE-D-23-38254R1

Dear Dr. Ekrikpo,

We’re pleased to inform you that your manuscript has been judged scientifically suitable for publication and will be formally accepted for publication once it meets all outstanding technical requirements.

Kind regards,

Ahmet Murt

Academic Editor

PLOS ONE

Additional Editor Comments (optional):

In this revised version of the manuscript, our impartial reviewers are satisfied with the explanations of the authors.

Reviewers' comments:

Reviewer's Responses to Questions

**Comments to the Author**

1. If the authors have adequately addressed your comments raised in a previous round of review and you feel that this manuscript is now acceptable for publication, you may indicate that here to bypass the “Comments to the Author” section, enter your conflict of interest statement in the “Confidential to Editor” section, and submit your "Accept" recommendation.

Reviewer #1: All comments have been addressed

Reviewer #2: All comments have been addressed

2. Is the manuscript technically sound, and do the data support the conclusions?

Reviewer #1: Yes

Reviewer #2: Yes

3. Has the statistical analysis been performed appropriately and rigorously? 

Reviewer #1: Yes

Reviewer #2: Yes

4. Have the authors made all data underlying the findings in their manuscript fully available?

Reviewer #1: Yes

Reviewer #2: Yes

5. Is the manuscript presented in an intelligible fashion and written in standard English?

Reviewer #1: Yes

Reviewer #2: Yes

6. Review Comments to the Author

Reviewer #1: (No Response)

Reviewer #2: With this revised version, authors have clarified all of my concerns. I have no further comments for this round of review.

7. PLOS authors have the option to publish the peer review history of their article (what does this mean?). If published, this will include your full peer review and any attached files.

Reviewer #1: No

Reviewer #2: No

---

## [Editor Report · Acceptance letter]

21 May 2024

PONE-D-23-38254R1 

PLOS ONE

Dear Dr. Ekrikpo, 

I'm pleased to inform you that your manuscript has been deemed suitable for publication in PLOS ONE. Congratulations! Your manuscript is now being handed over to our production team.

Kind regards, 

on behalf of

Dr. Ahmet Murt 

Academic Editor

PLOS ONE